# The LiverTox Paradox-Gaps between Promised Data and Reality Check

**DOI:** 10.3390/diagnostics11101754

**Published:** 2021-09-24

**Authors:** Rolf Teschke, Gaby Danan

**Affiliations:** 1Department of Internal Medicine II, Division of Gastroenterology and Hepatology, Klinikum Hanau, D-63450 Hanau, Germany; 2Academic Teaching Hospital of the Medical Faculty, Goethe University Frankfurt/Main, D-60590 Frankfurt/Main, Germany; 3Pharmacovigilance Consultancy, F-75020 Paris, France; gaby.danan@gmail.com

**Keywords:** iDILI, idiosyncratic drug-induced liver injury, Roussel Uclaf Causality Assessment Method (RUCAM), DILI database case quality, LiverTox

## Abstract

The LiverTox database compiles cases of idiosyncratic drug-induced liver injury (iDILI) with the promised aims to help identify hepatotoxicants and provide evidence-based information on iDILI. Weaknesses of this approach include case selection merely based on published case number and not on a strong causality assessment method such as the Roussel Uclaf Causality Assessment Method (RUCAM). The aim of this analysis was to find out whether the promised aims have been achieved by comparison of current iDILI case data with those promised in 2012 in LiverTox. First, the LiverTox criteria of likelihood categories applied to iDILI cases were analyzed regarding robustness. Second, the quality was analyzed in LiverTox cases caused by 46 selected drugs implicated in iDILI. LiverTox included iDILI cases of insufficient quality because most promised details were not fulfilled: (1) Standard liver injury definition; (2) incomplete narratives or inaccurate for alternative causes; and (3) not a single case was assessed for causality with RUCAM, as promised. Instead, causality was arbitrarily judged on the iDILI case number presented in published reports with the same drug. All of these issues characterize the paradox of LiverTox, requiring changes in the method to improve data quality and database reliability. In conclusion, establishing LiverTox is recognized as a valuable effort, but the paradox due to weaknesses between promised data quality and actual data must be settled by substantial improvements, including, for instance, clear definition and identification of iDILI cases after evaluation with RUCAM to establish a robust causality grading.

## 1. Introduction

The U.S. LiverTox database and website containing preferentially idiosyncratic drug-induced liver injury (iDILI) cases became available online in April 2012 and was published in March 2013 [1]. This new approach was much appreciated due to the promising intention to provide not only accurate but also complete summary information on the characteristics of clinical liver injury for each drug, along with an exhaustive and annotated reference compilation. Expectations were high because the website was created by the U.S. National Library of Medicine (NLM) and annotated by the U.S. National Institute of Diabetes and Digestive and Kidney Diseases (NIDDK) staff [1]. The initial website version was then replaced by a new version [2]. The data of iDILI cases were said to be collected from various sources, including clinicians submitting case reports [1]. The website of LiverTox consecutively produced a computer-generated history, associated with a table of laboratory data, and a graphic display of clinical details, which included calculations of latency, time to recovery, severity, and causal relationship by applying the scores of RUCAM (Roussel Uclaf Causality Assessment Method), with various steps, thus ensuring data completeness and high-quality data [1]. Additional details were provided and seemed overall promising. As a result, expectations among iDILI experts were high for using the database cases as reference cases for clinical and study purposes. However, expectations were only partially met due to problems such as those related to causality assessment, acknowledged by one of the initiators of the LiverTox database and website [3]. The critical view was in line with views of other groups criticizing shortcomings of the LiverTox initiative [3,4,5,6,7,8,9,10]. It seemed that a gap emerged between the promise to provide valid case data of high quality and the presented case data. Most cases are of insufficient quality, especially in terms of a minimum information and a lack evaluation of a causal relationship by applying a strong method such as the RUCAM, a real paradox with a clinical and research impact requiring additional evaluation. 

The present report analyzed the data quality of selected iDILI cases included in the LiverTox database. It turns out that in a significant number of cases, the presented data are not of sufficient quality to be used in a clinical or research setting. Suggestions of improvement are made, enforcing the application of a strong causality assessment method (CAM) such as the RUCAM in retained cases, whatever the data source. 

## 2. Materials and Methods 

A comparison was made between data currently included in the LiverTox database and on its website [2] and the aims or promises provided at LiverTox implementation [1]. First, the LiverTox criteria of likelihood categories applied to iDILI cases were analyzed regarding causality assessment. Second, the data quality was analyzed in iDILI cases selected from LiverTox, which presents on its website all drugs implicated in iDILI as blocks in alphabetic order from A to Z [2]. From each alphabetical block, the first listed single drug was selected for analysis, excluding groups of several drugs or nondrug products such as herbs. This led to 23 drugs from Abacavir to Zafirlukast implicated in iDILI cases. In addition, and to be on the safe side, a second list of 23 drugs from each alphabetical drug block was analyzed. Finally, in order to provide for LiverTox appropriate proposals to improve the data quality of iDILI cases, additional reports were sought to broaden the discussion, starting with a few reports [3,4,5,6,7,8,9,10,11,12,13,14,15,16,17,18,19,20,21,22,23,24,25]. 

## 3. Results

### 3.1. LiverTox Criteria of Likelihood Category 

In the LiverTox database, the causality grading is described in seven likelihood categories applied to iDILI cases [3,4]. In other sections of the database, these likelihood categories are termed likelihood “scores,” an inappropriate expression that should no longer be used [2]. Indeed, scores are usually achieved by adding individual scores attributed to specific items in the frame of an algorithm, which is preferentially adherent to Artificial Intelligence (AI) principles [11], conditions that do not apply to LiverTox [1,2,3,4]. Apart from the scoring issue, another problem is the inaccurate definition of the causality categories (Table 1).

Contrary to the promised application of RUCAM to assess the causality of iDILI cases in LiverTox as proposed in 2013 [1], this was not carried out for unknown reasons [2]. Early support for RUCAM was provided in 2011 by an expert group of scientists from various countries, including the U.S., stating that the causality of iDILI cases should be assessed with RUCAM [12], considering that approximately half of the cases were misdiagnosed [13,14]. Subsequently, the utility of RUCAM was confirmed in 2020 by an expert review on selected highlights of iDILI, and it was outlined that only iDILI cases assessed with RUCAM should be discussed [9]. In the same year, the RUCAM publications of 1993 by their founding authors [15,16] was one of the topics discussed in a scientometric study focusing on the worldwide knowledge mapping of liver injury caused by drugs, as outlined in a publication by Chinese experts [17]. 

Rather than applying RUCAM as promised in 2013 [1], LiverTox used another approach to assess causality [3,4], not involving RUCAM, as confirmed in 2021 [18], but arbitrarily classifying the iDILI cases of the LiverTox website into seven categories of likelihood, whereby the inclusion criterion was the number of reports retrieved from published studies (Table 1) [2,3,4]. In other words, with an increasing number of published reports, the causality grading of iDILI cases moved to higher causality levels [3,4]. This approach of causality assessment has never been validated and could lead to mistakes and clinical errors. Under these evaluating conditions, among 671 drugs, only 53% were classified as likely causing iDILI, considering the reports in the literature, while 47% were just based on an expert opinion lacking supportive evidence in the literature by previous reports [3,4], as critically discussed [6]. It was also outlined that although in LiverTox a thorough literature search had been approached, it was not attempted to assess the quality of the published reports or to evaluate the causality of liver injury [3]. Other analyses of the iDILI cases of the database found that approximately one half of drugs reviewed met the criteria for causing, or being suspected as causing, iDILI [6,9]. No efforts had been made to improve case data quality [18]. Overall, it seems that quality aspects remain a crucial issue of LiverTox.

Critical situations emerged when studies used iDILI cases retrieved from the LiverTox database, which may call the study results into question. For instance, an association between iDILI and daily dose, liver metabolism, and lipophilicity has been suggested, but proposals were based not on own valid iDILI cases rather than on cases uncritically retrieved from a variety of drug and DILI databases, including LiverTox [19,20,21]. Such an approach was critically discussed [22] in support of the statement that the proposed drug characteristics are not able to predict iDILI with high confidence, associated by a caveat note [23]. It is obvious that this controversy around risk factors of iDILI is fairly limited to FDA scientists [19,20,21,22,23], an interesting constellation calling for internal solutions [8,10,22].

Long before LiverTox was presented to the scientific community [1], national iDILI registries from Spain [24] and Sweden [25] successfully used RUCAM [15,16] for their cases, viewed as pioneering work and early trust in RUCAM [7]. As a sign of appreciation, experts discussed only RUCAM-based iDILI and HILI (herb-induced liver injury) cases, rather than cases of the LiverTox database, which were not assessed with RUCAM [9]. It appears that LiverTox is still far behind mainstream approaches, causing concern about data quality [1,2,8,10,18]. 

### 3.2. Quality Assessment of Selected iDILI Cases

Several hundred iDILI cases are included on the LiverTox website, but the exact number remains unknown because liver injury cases by nondrugs such as herbs, herbal traditional medicines, common herbal products, and so called dietary supplements are also listed [1,2,3,4,10]. LiverTox data quality was evaluated only in a portion of these iDILI cases [1,2,3,4,5,18]. In the first list, 23 drugs implicated in iDILI cases of LiverTox were available for this analysis (Table 2).

For each drug, the LiverTox based likelihood categories from A to E were added (Table 2), as retrieved from the website (Table 1) [2]. Where available, additional data were derived from the website for each listed drug and added to the drug, associated with a commentary if applicable (Table 2). No case details were provided for 13/23 (56.5%) drugs implicated in iDILI, and a commentary was, therefore, not possible for these 13 cases (Table 2). In 4/13 cases lacking details, a “probable” likelihood was attributed based on more than 50 published case reports for these drugs (Table 2). Similar insufficient data were obtained for the second list, consisting again of 23 drugs implicated in iDILI (data not shown). Among 10/23 (43.5%) iDILI cases of the first list, incomplete data were presented, not allowing for a case evaluation (Table 2). Overall, in association with insufficient approaches of causality assessment (Table 1), this compilation confirms poor data quality (Table 2). 

Published first in 2005, selected registries reported iDILI cases commonly with a causality grading of “probable” or “highly probable” following assessment with RUCAM that allows for correct case features description [10,24,25]. On the contrary, since 2013 LiverTox has included 60.9% of selected cases with a causality grading of possible or lower, and only 39.1% of the cases can be found in the category of “highly probable,” “highly likely,” or “probable” causality categories (Table 3) [2], calling for a modified approach. 

As compared to LiverTox, iDILI cases of a better quality are available from a large worldwide study on 81,856 published iDILI cases, all assessed with RUCAM [26], either in the original version [15,16] or the updated version published 2016 [27], which should now preferentially be used [28,29]. It is suggested that the scientists involved in LiverTox maintenance are encouraged to search for RUCAM-based iDILI cases to be included in LiverTox.

### 3.3. LiverTox Paradox Based on Gaps 

There was much hope among members of the iDILI community when LiverTox entered the challenging field of including valid iDILI cases in a new database and on a website [1,2]. Soon, however, it became evident that the applied CAM based on an expert opinion process and the quality of the cases did not meet the expectations (Table 1, Table 2 and Table 3) [2,3,4,5,6,7,8,9,10]. As a result, the gaps created a paradox: The promise of providing valid case data versus presenting cases of insufficient quality. The identified gaps relate to the completeness and accuracy of the case details, evidence-based features, and causality assessment (Table 4). 

### 3.4. Suggestions for Improvement 

Since the time of LiverTox implementation, a variety of suggestions have been made to ensure some degree of high quality of the iDILI cases to be included [1]. However, the diagnostic causality approach and data presentation of LiverTox remain outside of mainstream opinion (Table 1 and Table 2). These approaches are not acceptable because they are subjective, not transparent, not structured, not based on strict working procedures leading to variable results, not excluding alternative causes, not validated, and not based on an element scoring allowing for a final causality grading. The results, as presented by LiverTox, are disappointing (Table 1, Table 2, Table 3 and Table 4) and have become a matter of debate [2,3,4,5,6,7,8,9,10]. A new approach is now required to improve the quality of the LiverTox database. To achieve this, some proposals are made (Table 5).

### 3.5. Use of RUCAM 

LiverTox authors should now get started on using RUCAM prospectively for iDILI cases, in line with the promise in 2013 [1] and according to the proposal of extending our knowledge by increasing population analysis with prospective causality evaluation using a scoring approach [30]. RUCAM is appreciated throughout the world [6,7,8,9,12,17,24,25,26,27,31,32] as a structured, transparent, user friendly, objective, and quantitative diagnostic algorithm [27] according to AI recommendations [11]. In addition, RUCAM is conceptualized as a diagnostic method specific for hepatic injury caused by drugs and herbs [27] and rarely needs expert assistance, except perhaps in special populations such as those with hepatitis [33]. As an overview, some specifics of RUCAM are provided in a condensed form (Table 6).

Causality assessment with RUCAM used by an independent team of experts was reproducible within clinical acceptable limits [15,16]. In addition, validation of RUCAM was achieved with cases considering, among other features, a positive test of rechallenge [16]. No validation method was used by authors of any other CAM [34]. The authors of LiverTox should also benefit from the experience of an independent group not involved in any CAM creation, which reported a low variability of intra-observer features without disagreement in the evaluation of iDILI cases when using RUCAM [25]. 

RUCAM characteristics are at variance with those of other CAMs that are not specifically prepared to evaluate injury of the liver [34], are devoid of element specification, or lack a scoring system [35]. As a result, most of them are not appropriate for application to causality assessment in liver injury cases because they are not liver-specific, are subjective rather than objective as based on variables, are often divergent in terms of the opinions of the assessing scientists of physicians, are not validated with a gold standard such as a positive test of unintentional re-exposure, and finally do not present reproducibility of causality levels derived from scored key elements [27,34]. Known from other well-established methods in medicine, background noise is not unexpected to be provided preferentially by peers, who have never published before a validated diagnostic algorithm of iDILI suitable for worldwide application. In this context, several unsuccessful attempts by authors to add, modify, or delete key elements or to upgrade or downgrade scores were frustrating in the past. Overall, these less-convincing approaches reduced the user-friendly handling of the method, making RUCAM application more complicated without chances of validation. Not unexpected, the data were not published. A discussion is warranted with respect to the diagnostic biomarkers outlined in several critical publications [8,9,36]. They are commonly not based on iDILI cases assessed for causality with RUCAM [36], most of them lost support by the EMA (European Medicines Agency) and the U.S. FDA due to misconducted research as detailed elsewhere [9], and they certainly cannot replace RUCAM [8]. 

## 4. Discussion

Around one dozen iDILI databases are publicly available with features and limitations well described, whereby the LiverTox database ranks among the major ones and needs to be further improved [5]. In this context, it has been argued that the interface of LiverTox does not allow for an intuitive approach, impairing for users access to data, and the search request form was found to be limited. Despite these minor shortcomings, there have been many more discussions about the data quality of cases included in the LiverTox database [4,5,6,7,8,9,10]. Currently, there are criteria problems of causality attribution by not using a robust CAM such as RUCAM, as promised (Table 1), and insufficient quality of iDILI cases (Table 2), not allowing for appropriate use of included cases by physicians. Difficult to reconcile is the gap between the initially promised excellence of case data quality and the finally presented quality (Table 4). Despite some shortcomings, case quality is much better in national DILI registries using the prospective RUCAM, with a few exemptions [10]. This suggests their inclusion in the LiverTox database. 

This analysis may have some limitations; for example, only 52 drugs were randomly included. As with any well-working method in medicine, some background noise is expected for various reasons, especially from scientists, who have never established a robust algorithm such as RUCAM. In this context, several unsuccessful attempts by others to add, modify, or delete elements or to upgrade or downgrade scores were frustrating, reducing the user-friendly handling of the method, making the method more complex and not validated. The data were, as expected, not published. In fact, and as in real-life situations, a well running method such as RUCAM should not be changed unless major improvements are expected. If a robust diagnostic biomarker emerges, derived from RUCAM-based DILI and HILI cases, its inclusion in RUCAM is not recommended unless a full and new validation process is carried out. Rather, it should be used in parallel to the updated RUCAM. Finally, causality assessment based on expert opinion is not recommended for various reasons and remains debated due to major shortcomings [27,34,35], a view supported by the above analysis on LiverTox cases.

## 5. Conclusions

Among the various iDILI databases or websites, the implementation of the U.S. LiverTox database in 2013 was highly appreciated as a new data source of drugs causing iDILI. However, its use by clinicians can be limited by problems of data quality, including missing detailed narratives and evidence-based case features, a lack of standard definitions such as liver injury, and the failure in all cases to use a strong method for evaluating causality such as RUCAM, although use of this method was initially promised by LiverTox. As it presently stands, there are major gaps between the promised details and the provided facts, as shown in this analysis. These gaps were unexpected and considered as a paradox, calling now for improvements. In essence, LiverTox will gain appreciation if iDILI cases of better quality are included. Courage is now required.

## Figures and Tables

**Table 1 diagnostics-11-01754-t001:** Categories of the likelihood of iDILI in LiverTox.

LiverTox Likelihood Categories of iDILI Cases	Criteria of Likelihood Categories Applied to iDILI Cases Included in the LiverTox Database
Category A: Highly probable	The drug is well known, well described, and well reported to cause either direct or idiosyncratic liver injury, and has a characteristic signature; more than 50 cases, including case series, have been described.
Category B: Highly likely	The drug is reported and known or highly likely to cause idiosyncratic liver injury and has a characteristic signature; between 12 and 50 cases, including small case series, have been described.
Category C: Probable	The drug is probably linked to idiosyncratic liver injury, but has been reported uncommonly and no characteristic signature has been identified; the number of identified cases is less than 12 without significant case series.
Category D: Possible	Single case reports have appeared, implicating the drug, but fewer than three cases have been reported in the literature, no characteristic signature has been identified, and the case reports may not have been very convincing; thus, the agent can only be said to be a possible hepatotoxin and only a rare cause of liver injury.
Category E: Unlikely	Despite extensive use, no evidence that the drug has caused liver injury. Single case reports may have been published, but they were largely unconvincing. The agent is not believed or is unlikely to cause liver injury.
Category E: Unproven	The drug is suspected to be capable of causing liver injury or idiosyncratic acute liver injury, but there have been no convincing cases in the medical literature. In some situations, cases of acute liver injury have been reported to regulatory agencies or mentioned in large clinical studies of the drug, but the specifics and details supportive of causality assessment are not available. The agent is unproven but suspected to cause liver injury.
Category X: Not assessed	Finally, for medications recently introduced into or rarely used in clinical medicine, there may be inadequate information on the risks of developing liver injury to place it in any of the five categories, and the category is characterized as “unknown.”

Listed causality gradings were arbitrary and taken word-by-word from previous publications [2,3,4]. Abbreviations: iDILI, idiosyncratic drug-induced liver injury.

**Table 2 diagnostics-11-01754-t002:** Selected reports with questionable causality assessed by LiverTox.

Drug	LiverTox Category of Case Likelihood	LiverTox iDILI Case Details, Confounding Variables, Alternative Causes and Comments
Abacavir	Category C: Probable	HEV, HSV, and VZV infections were not excluded. Comedication with nevirapine, lamivudine, lopinavir. Commentary: Consider better as alternative causes: HEV, HSV, VZV infection, or comedication.
Baclofen	Category D: Possible	No details of a specific case provided. No commentary.
Cabazitaxel	Category E: Unproven	No details of a specific case provided. No commentary.
Dabigatran	Category E: Unproven	No details of a specific case provided. No commentary.
Eculizumab	Category D: Possible	A specific case presented without details of exclusion of alternative causes. Small case series without details provided. No commentary.
Famciclovir	Category E: Unlikely	No details of a specific case provided. No commentary.
Gabapentin	Category C: Probable	No details of a specific case provided. No commentary.
Haloperidol	Category B: Highly likely	No details of a specific case provided. No commentary.
Ibalizumab	Category E: Unlikely	No details of a specific case provided. No commentary.
Ketamine	Category B: Highly likely	Single case, no acute DILI because ketamine was inhaled for 9 months. Tests for HAV, HBV, and HCV were unremarkable, as were those for autoantibodies and Wilson disease. Liver histology was suggestive of primary sclerosing cholangitis (PSC). Commentary: Rather than acute iDILI, PSC is the most likely diagnosis.
Labetalol	Category C: Probable	Single case presented with lethal outcome. Patient was negative for hepatitis A and B. No diagnosis was made, and the patient again received at two different occasions of labetalol, leading to lethal ALF. Commentary: Differential diagnosis of ALF poorly assessed.
Macitentan	Category E: Unlikely	No details of a specific case provided. No commentary.
Nabilone	Category E: NA	No details of a specific case provided. No commentary.
Obeticholic acid	Category B: Highly likely	Single case of a patient with PSC lacking exclusion of alternative causes. Commentary: Case is best seen as exacerbation of PSC rather than as acute iDILI.
Paclitaxel	Category D: Possible	Single case of a severely ill patient, with previous pelvic radiation and now carboplatin comedication, who experienced a severe hypersensitivity reaction and increased liver tests without assessing alternative causes. Commentary: Poorly documented case of unclear iDILI.
Quazepam	Category E: Unlikely	No details of a specific case provided. No commentary.
Rabeprazole	Category D: Possible	No details of a specific case provided. No commentary.
Safinamide	Category E: Unlikely	No details of a specific case provided. No commentary.
Tacrine	Category A: Highly probable	Single case presented, vague exclusion of alternative causes. Commentary: Poorly documented case.
Ursodiol	Category D: Possible	No details of a specific case provided. No commentary.
Valacyclovir	Category D: Possible	Single case presented of a patient with shingles; tests for hepatitis A, B, and C were negative, as were autoantibodies. Comedication with acetaminophen. Specific note: The possibility of varicella zoster-induced hepatitis should also be considered. Commentary: Increased values of ALT and ALP are best explained by the liver involvement of varicella zoster virus infection and not by iDILI.
Warfarin	Category C: Probable	No details of a specific case provided. No commentary.
Zafirlukast	Category C: Probable	Case 1: Patient was described as having no risk factors for viral hepatitis. Test for hepatitis A, B, and C were negative, as were autoantibodies, and other parameters to exclude alternative causes were not presented. Commentary: Insufficiently documented case, not allowing for a valid diagnosis.Case 2: Patient was reported as having no history of exposure to viral hepatitis, but details of hepatitis exclusion were not provided. Positive results of unintentional reexposure were described without presenting applied criteria. Commentary: Poorly documented case.

Listed details were retrieved from an earlier publication [2]. Abbreviations: ALF, acute liver failure; ALP, alkaline phosphatase; ALT, alanine aminotransferase; iDILI, idiosyncratic drug-induced liver injury; NA, not available; PSC, primary sclerosing cholangitis.

**Table 3 diagnostics-11-01754-t003:** Distribution of causality gradings among the selected iDILI cases of the LiverTox database.

Causality Grading	iDILI Cases (*n*)	iDILI Cases (%)
Highly probable	1	4.4
Highly likely	3	13.0
Probable	5	21.7
Possible	6	26.1
Unlikely	5	21.7
Unproven	2	8.7
Not assessed	1	4.4

Data were collected from Table 2. Abbreviations: iDILI, idiosyncratic drug-induced liver injury.

**Table 4 diagnostics-11-01754-t004:** LiverTox paradox: Promised data versus presented data.

Promised Data	Presented Data and Gaps
Cases of iDILI with RUCAM scores [1].	Evidence is missing that RUCAM was ever used in any iDILI case included in the LiverTox database or presented on the website [2].
A complete and accurate summary of information about the clinical features of liver injury for each drug [1].	Clinical summaries were incomplete due to a lack of a diagnostic algorithm such as RUCAM to assess causality [2]. Instead, causality gradings were arbitrarily published considering the number of published case reports.
A website with comprehensive and evidence-based detailed information on iDILI cases [1].	Information was incomplete and not evidence-based, because the causality was not assessed with a robust method such as RUCAM [2] that would have assessed the exclusion of alternative causes.
A separate section on detailed information about formal CAMs such as RUCAM [1].	The section is not up to date. References are, for instance, to 2 reports of RUCAM in 1993 [15,16] and not actualized in 2016 [2] with the updated version [27], followed by additional information [28,29].
Providing standardized definitions of terms used [1].	Standard criteria of liver injury such as ALT higher than 5 × ULN and/or ALP higher than 2 × ULN [12,27] are not presented [2].

Details were derived from a published report [1], the LiverTox website [2], and Table 2. Abbreviations: ALP, alkaline phosphatase; ALT, alanine aminotransferase; CAMs, causality assessment methods; iDILI, idiosyncratic drug-induced liver injury; RUCAM, Roussel Uclaf Causality Assessment Method.

**Table 5 diagnostics-11-01754-t005:** Proposals to improve the data quality in LiverTox.

Proposals
1. Clinicians, as potential authors of RUCAM-based iDILI case reports, should be encouraged to submit their case reports directly to LiverTox. 2. Additional RUCAM-based iDILI cases should be retrieved from existing iDILI registries of various countries or regions [10], including, for example, Sweden [25], Iceland [30], Spain [24], and Latin America [31], all of which collect iDILI cases using a prospective approach. 3. Other RUCAM-based iDILI cases should be selected from the 81,856 cases published from 1993 to mid-2020 [26]. 4. From now on, the prerequisites for iDILI cases to be included in the LiverTox database and website should be: ● Liver injury must be defined as ALT higher than 5 × ULN (upper limit of normal) and/or ALP higher than 2 × ULN [12,27]; ● An informative case narrative with complete diagnostic and clinical details [27]; ● The R ratio must be calculated based on ALT and ALP values in order to classify the case as hepatocellular injury or cholestatic/mixed liver injury [27]; ● The case should be assessed with the updated RUCAM [27], and the final score should be provided.

Abbreviations: ALP, alkaline phosphatase; ALT, alanine aminotransferase; iDILI, idiosyncratic drug-induced liver injury; RUCAM, Roussel Uclaf Causality Assessment Method; ULN, upper limit of normal.

**Table 6 diagnostics-11-01754-t006:** Ranges of scores for individual RUCAM elements in iDILI cases of hepatocellular injury or the cholestatic/mixed liver injury.

Elements Assessed by RUCAM	Scores of RUCAM for Hepatocellular Injury	Scores of RUCAM for the Cholestatic/ Mixed Liver Injury
● Time frame of latency period	From +1 to +2	From +1 to +2
● Time frame of dechallenge	From −2 to +3	From 0 to +2
● Recurrent ALT increase	−2	-
● Recurrent ALP increase	-	0
● Risk factors	0 or +1	0 or +1
● Individual comedication	From −3 to 0	From −3 to 0
● Search for individual alternative causes	From −3 to +2	From −3 to +2
● Verified exclusion of alternative causes	Requires individual scoring	
● Markers of HAV, HBV, HCV, and HEV		
● Markers of CMV, EBV, HSV, and VZV		
● Evaluation of cardiac hepatopathy		
● Liver and biliary tract imaging		
● Doppler sonography of liver vessels		
● Prior known hepatotoxicity	From 0 to +2	From 0 to +2
● Unintentional reexposure	From −2 to +3	From −2 to +3

Presented in condensed form, the above listed details were retrieved from an earlier publication on the updated RUCAM to be used for causality evaluation [27]. Additional information of each criterion and score is given in the RUCAM worksheet [28]. Total score and resulting causality grading: ≤0, excluded; 1–2, unlikely; 3–5, possible; 6–8, probable; ≥9, highly probable. Abbreviations: ALP, alkaline phosphatase; ALT, alanine aminotransferase; CMV, cytomegalovirus; EBV, Epstein–Barr virus; HAV, hepatitis A virus; HBV, hepatitis B virus; HCV, hepatitis C virus; HEV, hepatitis E virus; HSV, herpes simplex virus; RUCAM, Roussel Uclaf Causality Assessment Method; VZV, varicella zoster virus.

## Data Availability

Data supporting the results of the analysis can be retrived from the following sources: LiverTox. Clinical and Research Information on Drug-Induced Liver Injury. Bethesda (MD): National Institute of Diabetes and Digestive and Kidney Diseases; 2012-. LiverTox Database. Updated: 9 February 2021. Available at: https://www.ncbi.nlm.nih.gov/books/NBK547852/ accessed on 19 May 2021; Hoofnagle JH. Relaunched LiverTox remains important resource to diagnosis DILI. Available at: https://www.aasldnews.org/relaunched-livertox-remains-important-resource-to-diagnosis-dili/ accessed on 19 May 2021.

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
