# Peer review of "The LiverTox Paradox-Gaps between Promised Data and Reality Check"

_diagnostics, 2021, doi:10.3390/diagnostics11101754_

Round 1
Reviewer 1 Report
Drs Teschke and Danan did a study to determine the quality of iDILI case data reported in LiverTox. They did an analysis of (26 plus 26) random cases from each 71 alphabetical drug block. All over the article, they comment different points of view, with suggestions to improve the LiverTox web site. The article is well written, with a really interesting analysis followed by proposals. Only few minor comments should be discussed:
Minor Comments:
1) The RUCAM system, is well accepted but some problems have been discussed in the scientific communities, such as this system is problematic for future studies of drug induced liver injury. Alternative methods, including modifying the RUCAM, developing drug specific instruments, or causality assessment based on expert opinion, may be more appropriate. Authors should comment on the other systems and why RUCAM is indeed the most appropriate, in a paragraph on the discussion section.
2) This analysis has some limitations, for example that only 46 drugs were randomly included. Authors should include a paragraph with the study limitations on the discussion section.
Author Response
Thank you for encouragement and proposals, all of them were included shown in red.
Reviewer 2 Report
Here, the authors evaluated the quality of the LiverTox database as it exists today, with reference to the original intent and plans for the database. It's a useful discussion to have. I have minor comments:
- The English requires editing for spelling and grammar. I found portions of the article difficult to understand due to the poor writing. Some basic examples: "Discussion" is misspelled "Discution" in the heading for the discussion section, "hepatoxin" is used instead of the proper term "hepatotoxicant" (toxin refers to naturally-occurring poisons, such as amatoxins and microcystin, while synthetic drugs, pesticides, etc. are called toxicants). Please revise carefully.
- 50% of the references (19/38) are the author's own past publications. Of course, when someone is dominant in a field, they will frequently have to cite themselves because they produce more than others. Nevertheless, I would ask the authors to review their references and ensure they are citing themselves appropriately and check to see if there are other papers they should cite as well.
Author Response

(The authors gave the same response as above.)

Round 2
Reviewer 1 Report
The authors have satisfactorily responded to all my questions and made the necessary changes to the manuscript.
Thank you very much.
Reviewer 2 Report
No further comments.